# Beyond the Ground Truth:
# Enhanced Supervision for Image Restoration

## Abstract

Deep learning-based image restoration has achieved significant success. However, when addressing real-world degradations, model performance is limited by the quality of ground-truth images in datasets due to practical constraints in data acquisition. To address this limitation, we propose a novel framework that enhances existing ground truth images to provide higher-quality supervision for real-world restoration. Our framework generates perceptually enhanced ground truth variants using super-resolution, and employs a conditional frequency mask generator to produce adaptive frequency masks. These masks guide the optimal fusion of frequency components from the original ground truth and its super-resolved variants to yield enhanced ground truth images. This frequency-domain mixup preserves the semantic consistency of the original content while selectively enriching perceptual details, preventing hallucinated artifacts that could compromise fidelity. The enhanced ground truth images are used to train a lightweight output refinement network that can be seamlessly integrated with existing restoration models. Extensive experiments demonstrate that our approach consistently improves the quality of restored images. We further validate the effectiveness of both supervision enhancement and output refinement through user studies. We will publicly release our code, enhanced images and model weights to support reproducibility.

## 1 Introduction

Image restoration has achieved remarkable progress through supervised training on paired low-quality and ground truth images using deep neural networks. Across various degradation types, a range of architectures (Zhang et al., 2017; Kupyn et al., 2018; Liang et al., 2021; Chen et al., 2022; Zamir et al., 2022; Guo et al., 2024b) and learning strategies (Lehtinen et al., 2018; Ulyanov et al., 2018; Yoo et al., 2020; Zhang et al., 2022; Wu et al., 2024a) have been proposed to align restored outputs closely with ground truth images. Recently, the focus has shifted toward improving perceptual quality of the restored outputs, leveraging advances in generative models to produce visually compelling results (Wang et al., 2024; Lin et al., 2024; Yu et al., 2024; Wu et al., 2024c).

Despite these advances, in real-world image restoration where acquiring ideal reference images is inherently difficult due to practical constraints in data acquisition, improving perceptual quality remains a significant challenge. Many existing datasets rely on indirect ways to construct ground truth images. For instance, in deblurring datasets (Nah et al., 2017; Shen et al., 2019; Nah et al., 2019), ground truth images are selected from video sequences, which often contain slight camera shake or object movements, limiting the image sharpness. Likewise, in denoising datasets (Nam et al., 2016; Abdelhamed et al., 2018; Xu et al., 2018), ground truth images are constructed by averaging multiple noisy captures, often resulting in blurred references. As a result, models trained on such suboptimal ground truth images inevitably tend to inherit those imperfections, limiting their ability to achieve high-quality restoration.

To address this limitation, we propose a novel supervision enhancement framework designed to improve the perceptual quality of suboptimal ground truth images. The proposed framework consists of two main components: (1) super-resolution using a one-step diffusion model to generate perceptually enhanced ground truth variants, and (2) frequency-domain mixup to produce the final enhanced ground truth images. For the frequency-domain mixup, we introduce a conditional frequency mask generator that adaptively produces masks to guide the optimal fusion of frequency components from

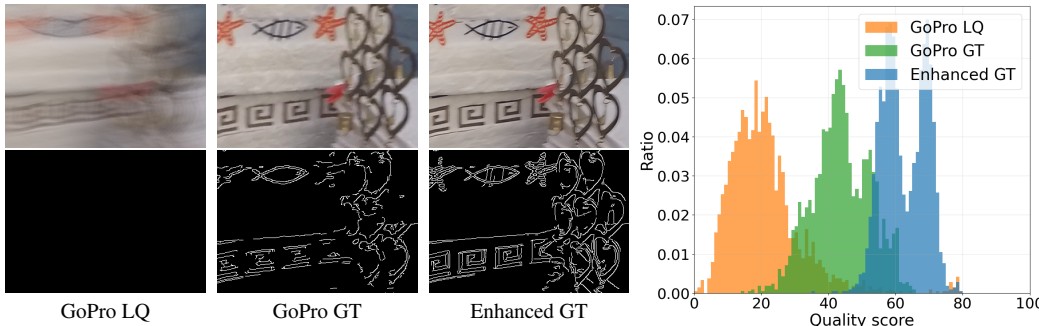

Figure 1: Images and their corresponding edge maps (left) and quality scores measured by KonIQ++ (Su et al., 2021) (right) for the GoPro (Nah et al., 2017) training set. Our enhanced ground truth images exhibit better sharpness and achieves higher quality scores.

the original ground truth image and its super-resolved variants. As illustrated in Figure 1, the resulting enhanced ground truth images provide clearer details and higher perceptual quality than original.

Building upon the enhanced ground truth images, we design a lightweight output refinement network that can be seamlessly integrated into a wide range of pretrained restoration models without requiring architectural changes or retraining. Experiments show that the refinement network consistently improves the quality of restored images, benefiting from the enhanced supervision provided by our framework. Moreover, the network exhibits strong robustness in out-of-distribution scenarios, effectively removing residual degradations that remain after initial restoration. User studies further confirm the superior quality of both the enhanced ground truth images and the refinement outputs.

In summary, our key contributions are organized as follows:

- We identify the limitations of conventional ground truth images as a critical bottleneck in real-world image restoration, and propose a supervision enhancement framework based on frequency-domain mixup of an original ground truth image and its super-resolved variants. This design preserves semantic fidelity while enriching perceptual details, resulting in more reliable supervisory signals.

- We introduce a lightweight refinement network that is trained solely on the original and enhanced ground truth images, requiring no additional annotations. The module is model-agnostic, seamlessly integrating with arbitrary restoration backbones without architectural modifications or retraining, and is empirically shown to be robust even under out-of-distribution degradations.

- We validate our approach through extensive experiments and user studies, demonstrating consistent improvements in both enhanced ground truth quality and restored image fidelity.

## 2 RELATED WORKS

With the rise of deep learning, traditional image restoration methods have largely been replaced by data-driven approaches trained on paired low-quality and ground truth images. A wide range of architectures has been proposed, including Convolutional Neural Networks (Dong et al., 2014; Zhang et al., 2017; Chen et al., 2022), Transformer-based models (Zamir et al., 2020; Liang et al., 2021; Zamir et al., 2022), Generative Adversarial Networks (Ledig et al., 2017; Kupyn et al., 2018), and more recently, state-space models such as Mamba (Guo et al., 2024b;a). These models have typically been trained to maximize metrics such as PSNR and SSIM, which quantify the pixel-wise similarity to the original ground truth images. While these approaches achieve high performance on standard benchmarks, their outputs often lack perceptual realism of high-quality images.

Recently, there is a growing interest in enhancing the perceptual quality of restored images. This has been particularly prominent in image super-resolution, demonstrating significant advancements in generating visually plausible high-frequency details (Xia et al., 2023; Delbracio & Milanfar, 2023; Wang et al., 2024; Lin et al., 2024; Yu et al., 2024; Wu et al., 2024c). This trend extends to broader restoration tasks, such as deblurring and denoising, where diffusion models have been leveraged to

enhance perceptual quality (Ohayon et al., 2021; Kawar et al., 2022; Luo et al., 2023; Zhu et al., 2023; Yue et al., 2024; Liu et al., 2025). While these methods effectively enhance perceptual quality, they often incur significant inference overhead and, more critically, risk hallucinating details or textures absent in the ground truth. In contrast, our approach aims to enhance perceptual quality while preserving the semantics of the original content by our novel frequency mixup strategy. To assess the perceptual quality of restored images, we employ a combination of deep-learning based image quality assessments (Ke et al., 2021; Yang et al., 2022; Chen et al., 2024a; Zhang et al., 2023), and emerging Vision-Language Model-based methods (Wu et al., 2025; Li et al., 2025).

## 3 SUPERVISION ENHANCEMENT FRAMEWORK

In this section, we introduce our supervision enhancement framework, which improves the perceptual quality of ground truth images in existing datasets to provide better supervision for image restoration tasks. The framework consists of two main components: (1) super-resolution using a one-step diffusion model to generate perceptually enhanced ground truth variants, and (2) combining these variants with the original ground truth image through frequency-domain mixup using masks generated by a conditional frequency mask generator. Figure 2 (a) illustrates an overview of our framework.

### 3.1 ENHANCING PERCEPTUAL QUALITY WITH IMAGE SUPER-RESOLUTION

Recent Image super-resolution (ISR) models have shown remarkable capability in improving perceptual quality. These models are trained using a combination of reconstruction and regularization losses, where the regularization term is crucial in learning natural image distributions and improving output quality. Typically, ISR models are trained to align the distribution of generated samples $q(\hat{x})$ with the distribution of high-quality real images $p(x_H)$, by minimizing the Kullback-Leibler divergence:

$$\mathcal{D}_{\mathrm{KL}}(q(\hat{x})||p(x_H)). \tag{1}$$

Typically, the distribution $p(x_H)$ is acquired from datasets with genuinely high-quality images, such as DIV2K (Agustsson & Timofte, 2017) or LSDIR (Li et al., 2023), or by leveraging the high-quality image manifold of large-scale pre-trained diffusion models. As a result, ISR models trained on this regularization effectively generate super-resolved outputs $\hat{x}$ of high perceptual quality.

Leveraging this capability, we adopt an one-step diffusion ISR model (Wu et al., 2024b) to enhance the suboptimal ground truth images. Specifically, each original ground truth image $I_0^{\mathrm{GT}}$ is first upsampled using bicubic interpolation with $N$ multiple scale factors.w The diffusion-based ISR model is then applied to these upsampled images, and the outputs are downsampled back to the original resolution, yielding a set of perceptually improved ground truth variants $\{I_i^{\mathrm{GT}}\}_{i=1}^N$.

### 3.2 FREQUENCY-DOMAIN MIXUP

Although image super-resolution (ISR) can enhance the perceptual quality of ground truth images, the generative nature of ISR models often introduces undesirable distortion in both semantics and photometric attributes. To alleviate these issues, we construct enhanced ground truth images by integrating the original ground truth with multiple super-resolved variants. A naive pixel-wise fusion in the spatial domain is problematic, since it effectively amounts to selecting or averaging pixel intensities across images, making it difficult to preserve high-level semantic structures and frequently introducing unrealistic artifacts. In contrast, we propose an adaptive frequency mixup, which provides fine-grained control by preserving essential low-frequency components in the original image while selectively incorporating perceptually richer high-frequency details from the super-resolved variants. This frequency-domain formulation is particularly suitable for image restoration tasks because it naturally harmonizes images with differing photometric characteristics, yielding more stable and visually coherent results than spatial-domain alternatives.

To facilitate optimal frequency fusion, we introduce a Conditional Frequency Mask Generator. As illustrated in the Figure 2 (b), given a set of input images $\{I_i^{\mathrm{GT}}\}_{i=0}^N$, where $i = 0$ denotes the original ground truth and $i = 1, \ldots, N$ denote its super-resolved variants, the mask generator outputs frequency masks $M_i$ by combining a set of predefined ring-shaped Gaussian basis masks $\{R_b\}_{b=1}^B$, and predicted coefficients for each basis.

(a) Supervision Enhancement Framework

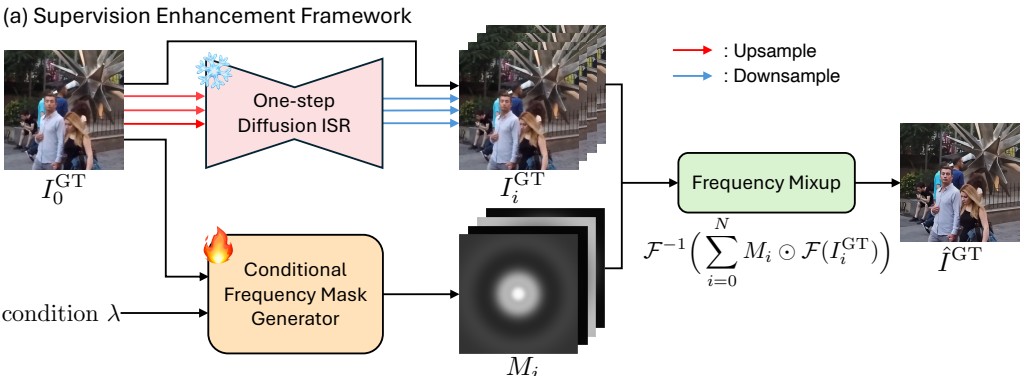

(b) Conditional Frequency Mask Generator

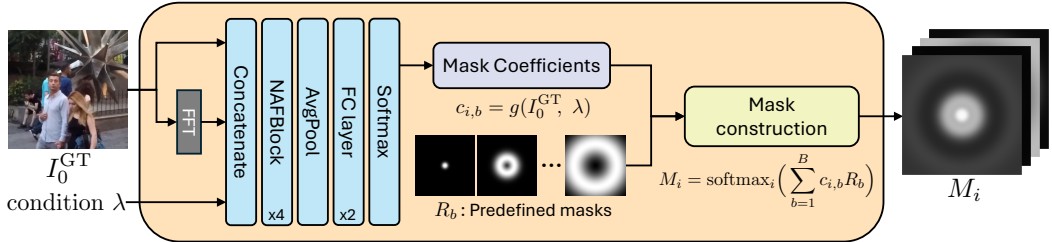

Figure 2: Overview of our framework. (a) The supervision enhancement framework produces enhanced ground truth images by fusing frequency components from the original ground truth $I_0^{\text{GT}}$ and its super-resolved variants $I_i^{\text{GT}}$ using adaptive frequency masks $M_i$. (b) The conditional frequency mask generator constructs $M_i$ by combining predefined masks $R_b$ weighted with predicted coefficients $c_{i,b}$, followed by a softmax function.

The design of our ring-shaped Gaussian basis masks is crucial for two reasons. First, the ring-shaped structure enables precise control from low to high frequencies in a band-wise manner. Second, the Gaussian shape ensures smooth transitions between frequencies, unlike discrete masks that introduce sharp boundaries, causing training instability and visual artifacts.

Specifically, each basis mask $R_b \in \mathbb{R}^{H \times W}$ is defined as:

$$(R_b)_{h,w} = \exp\left(-(d(h,w) - \mu_b)^2/2\sigma_b^2\right), \quad \text{for } 1 \le h \le H, \ 1 \le w \le W, \tag{2}$$

where $d(h,w)$ denotes the $\ell_2$-distance from the frequency-domain center (DC component), and $\mu_b, \sigma_b$ represent the Gaussian parameters of the $b$-th mask.

Given the original ground truth image $I_0^{\text{GT}}$ and the conditional parameter $\lambda$ that adjusts the weight between the original and its variants, a mask coefficient prediction network $g$ predicts coefficients $c_{i,b} \in \mathbb{R}$ as follows:

$$c_{i,b} = g(I_0^{\text{GT}}, \lambda). \tag{3}$$

Internally, $g$ augments the RGB input with its FFT representation, enabling joint use of spatial and frequency-domain information for mask coefficient prediction.

Then, the adaptive frequency masks $M_i$ are computed by combining these bases using predicted coefficients:

$$M_i = \text{softmax}_i\left(\sum_{b=1}^{B} c_{i,b} R_b\right), \tag{4}$$

where the softmax operation ensures masks sum to one, $\sum_{i=0}^{N} M_i(h,w) = 1, \forall(h,w)$.

Finally, the enhanced ground truth image $\hat{I}^{\text{GT}}$ is constructed by fusing frequency components of the original ground truth $I_0^{\text{GT}}$ and its super-resolved variants $\{I_i^{\text{GT}}\}_{i=1}^{N}$ through frequency-domain mixup:

$$\hat{I}^{\text{GT}} = \mathcal{F}^{-1}\left(\sum_{i=0}^{N} M_i \odot \mathcal{F}(I_i^{\text{GT}})\right), \tag{5}$$

where $\mathcal{F}$ and $\mathcal{F}^{-1}$ denote Fourier and inverse Fourier transforms, respectively, and $\odot$ represents element-wise multiplication.

### 3.3 OPTIMIZATION

To predict the mask coefficients for optimal frequency fusion, we train the network $g$, which predicts coefficients for each basis, with a composite loss that balances semantic integrity and perceptual quality.

The reconstruction loss is a $\ell_2$-loss that enforces consistency with the the original ground truth $I_0^{\mathrm{GT}}$:

$$\mathcal{L}_{\mathrm{recon}} = \|\hat{I}^{\mathrm{GT}} - I_0^{\mathrm{GT}}\|_2^2. \tag{6}$$

The perceptual loss is defined by a combination of multiple no-reference IQA metrics that evaluates perceptual quality of images (e.g., MUSIQ (Ke et al., 2021), MANIQA (Yang et al., 2022), TOPIQ (Chen et al., 2024a)), denoted as $\mathrm{IQA}_k(\cdot)$:

$$\mathcal{L}_{\mathrm{percep}} = -\sum_k \mathrm{IQA}_k(\hat{I}^{\mathrm{GT}}). \tag{7}$$

The final training loss combines these two terms, with their relative weights controlled by $\lambda \in [0, 1]$:

$$\mathcal{L} = (1 - \lambda)\mathcal{L}_{\mathrm{recon}} + \lambda\mathcal{L}_{\mathrm{percep}}. \tag{8}$$

## 4 OUTPUT REFINEMENT NETWORK

We demonstrate the effectiveness of our enhanced ground truth images by training a lightweight Output Refinement Network (ORNet) to improve the outputs of existing restoration models. While training a full restoration network from scratch using low-quality inputs and enhanced ground truth images is possible, we observe that many state-of-the-art models are already well-optimized for original ground truth images. Therefore, we propose an efficient strategy that builds on top of a fixed, pre-trained restoration model $R_\phi$. Specifically, we introduce a modular output refinement network $R_\theta$, which is trained to refine the output of $R_\phi$. The overall image restoration is formulated as:

$$\hat{I} = R_\theta(R_\phi(I^{\mathrm{LQ}}), \lambda), \tag{9}$$

where $I^{\mathrm{LQ}}$ is the low-quality input image, $\lambda$ is a parameter to control the level of perceptual enhancement, and $\hat{I}$ is the final restoration output.

Since the pre-trained image restoration model $R_\phi$ produces outputs close to the original ground truth (i.e., $R_\phi(I^{\mathrm{LQ}}) \approx I_0^{\mathrm{GT}}$), we train $R_\theta$ to map the $I_0^{\mathrm{GT}}$, which resembles the output of $R_\phi$, toward the enhanced ground truth $\hat{I}^{\mathrm{GT}}$ generated by our framework. The training objective for $R_\theta$ is given by:

$$\mathcal{L} = \|\hat{I} - \hat{I}^{\mathrm{GT}}\|_2^2 \approx \|R_\theta(I_0^{\mathrm{GT}}, \lambda) - \hat{I}^{\mathrm{GT}}\|_2^2. \tag{10}$$

This refinement strategy is model-agnostic, allowing it to be flexibly applied on top of various low-level vision models without architectural modifications. We demonstrate the effectiveness and versatility of this approach through extensive experiments.

## 5 EXPERIMENTS

In this section, we evaluate the perceptual quality of both the enhanced ground truth images, produced by our supervision enhancement framework, and the outputs of our refinement network, through comprehensive quantitative and qualitative experiments. In addition, we conduct user studies to assess the perceptual validity of both the enhanced ground truth and the refined outputs.

### 5.1 EXPERIMENTAL SETTINGS

**Implementation Details** For the supervision enhancement framework, we adopt OSEDiff (Wu et al., 2024b) as the super-resolution network. We generate three super-resolved ground truth variants

Table 1: Evaluation on the GoPro deblurring test set.

| Method | Perceptual Quality Metrics | | | | VLM-based Metrics | |
|---|---|---|---|---|---|---|
| | MUSIQ↑ | MANIQA↑ | TOPIQ↑ | LIQE↑ | VisualQuality-R1↑ | Q-Insight↑ |
| Restormer (Zamir et al., 2022) | 45.05 | 0.5265 | 0.3346 | 1.5264 | 4.1246 | 3.4000 |
| NAFNet (Chen et al., 2022) | 45.33 | 0.5346 | 0.3368 | 1.5542 | 4.1539 | 3.4262 |
| ResShift (Yue et al., 2024) | 44.30 | 0.4934 | 0.3127 | 1.4090 | 3.9105 | 3.3480 |
| IR-SDE (Luo et al., 2023) | 46.13 | 0.5336 | 0.3410 | 1.6140 | 3.9735 | 3.3619 |
| DiffIR (Xia et al., 2023) | 46.00 | 0.5366 | 0.3412 | 1.5820 | 4.1544 | 3.4269 |
| HI-Diff (Chen et al., 2024b) | 45.86 | 0.5337 | 0.3398 | 1.5576 | 4.1554 | 3.4207 |
| AdaRevD (Mao et al., 2024) | 45.49 | 0.5363 | 0.3393 | 1.5660 | 4.1737 | 3.4386 |
| + ORNet (**Ours**) | **64.25** | **0.5916** | **0.4880** | **2.4291** | **4.1952** | **3.5206** |
| FFTformer (Kong et al., 2023) | 46.47 | 0.5420 | 0.3456 | 1.6130 | 4.0942 | 3.4569 |
| + ORNet (**Ours**) | **64.57** | **0.5949** | **0.4924** | **2.4664** | **4.1995** | **3.5278** |

Table 2: Evaluation on the SIDD denoising test set.

| Method | Perceptual Quality Metrics | | | | VLM-based Metrics | |
|---|---|---|---|---|---|---|
| | MUSIQ↑ | MANIQA↑ | TOPIQ↑ | LIQE↑ | VisualQuality-R1↑ | Q-Insight↑ |
| AP-BSN (Lee et al., 2022) | 20.17 | 0.3613 | 0.1977 | 1.0556 | 1.0170 | 1.5496 |
| MIRNet-v2 (Zamir et al., 2020) | 22.18 | 0.3770 | 0.2402 | 1.1855 | 1.0484 | 1.6197 |
| Restormer (Zamir et al., 2022) | 22.55 | 0.3839 | 0.2439 | 1.2190 | 1.0653 | 1.6620 |
| Xformer (Chen et al., 2024b) | 22.57 | 0.3828 | 0.2472 | 1.2040 | 1.0759 | 1.6710 |
| + ORNet (**Ours**) | **35.68** | **0.4310** | **0.3710** | **1.9510** | **1.3228** | **2.1227** |
| NAFNet (Chen et al., 2022) | 22.73 | 0.3937 | 0.2458 | 1.2189 | 1.0826 | 1.7060 |
| + ORNet (**Ours**) | **35.87** | **0.4380** | **0.3776** | **1.9591** | **1.3513** | **2.1584** |

using scale factors of 2, 3, and 4. The number of predefined masks, B, for constructing the final mask is set to 25. Visualizations of these basis masks are provided in the Appendix A.1. Both the mask coefficient prediction network $g$ and the output refinement network (ORNet) are built using NAFBlocks, following the architectural design of NAFNet (Chen et al., 2022). Specifically, $g$ consists of 4 NAFBlocks and 2 FC layers, while ORNet is built as a U-Net architecture of 4 encoder blocks, 1 middle block, and 4 decoder blocks. $\lambda$ is set to 0.3 during evaluation. Additional evaluation results for different $\lambda$ values are presented in the Appendix.

**Training Details**  We train two core networks: the mask coefficient prediction network $g$ and ORNet, using a combined dataset of GoPro (Nah et al., 2017) and SIDD (Abdelhamed et al., 2018). Both $g$ and ORNet are trained for $100K$ iterations with a batch size of 8, using random $512 \times 512$ crops. AdamW (Loshchilov & Hutter, 2019) optimizer with cosine annealing learning rate scheduler is used. The initial learning rate is $1 \times 10^{-4}$ for $g$, and $3 \times 10^{-4}$ for ORNet. The parameter $\lambda$ is uniformly sampled from $[0, 1]$ during training to support learning of diverse enhancement levels.

**Evaluation Setup**  We evaluate under two regimes, in-distribution (ID) and out-of-distribution (OOD). The ID regime corresponds to standard restoration setups with their provided GT images, while the OOD regime is constructed by applying additional synthetic degradations (e.g., blur, noise) to test robustness. In ID, the GT images are themselves suboptimal. Reference-based metrics such as PSNR, SSIM, and LPIPS (Zhang et al., 2018) measure pixel- or feature-level similarity to the given GTs; however, when the GTs themselves are imperfect, these scores no longer provide a valid assessment of true restoration quality. For example, a higher PSNR against those suboptimal GTs does not necessarily indicate better restoration, and vice versa. Therefore, for experiments under ID regime, we report only no-reference perceptual metrics, MUSIQ (Ke et al., 2021), MANIQA (Yang et al., 2022), TOPIQ (Chen et al., 2024a), LIQE (Zhang et al., 2023), together with two recent VLM-based IQA measures (VisualQuality-R1 (Wu et al., 2025), Q-Insight (Li et al., 2025)), which better reflect human perception. In OOD, restored outputs are often far worse than even the original GT. Here the GT, though imperfect, serves as a valid reference for fidelity. Thus, we complement the perceptual metrics with reference-based measures (PSNR, SSIM, LPIPS (Zhang et al., 2018)) computed against both the original and enhanced GTs, providing a comprehensive view of fidelity as well as perceptual quality.

Table 3: Evaluation on an OOD environment, where an additional Gaussian blur ($\sigma = 2.5$) is applied to the blurry input images of the GoPro test set.

| Method | Original GT | | | Enhanced GT | | | Perceptual Quality Metrics | | | |
|---|---|---|---|---|---|---|---|---|---|---|
| | PSNR↑ | SSIM↑ | LPIPS↓ | PSNR↑ | SSIM↑ | LPIPS↓ | MUSIQ↑ | MANIQA↑ | TOPIQ↑ | LIQE↑ |
| FFTFormer (Kong et al., 2023) | 24.56 | 0.7532 | 0.4714 | 23.68 | 0.7224 | 0.5441 | 22.3812 | 0.2284 | 0.1832 | 1.0108 |
| +ORNet (**Ours**) | **24.58** | **0.7670** | **0.3429** | **23.81** | **0.7405** | **0.3777** | **42.9131** | **0.2638** | **0.2646** | **1.0656** |

Table 4: Evaluation on an OOD environment, where an additional white noise ($\sigma = 9$) is applied to the blurry input images of the GoPro test set.

| Method | Original GT | | | Enhanced GT | | | Perceptual Quality Metrics | | | |
|---|---|---|---|---|---|---|---|---|---|---|
| | PSNR↑ | SSIM↑ | LPIPS↓ | PSNR↑ | SSIM↑ | LPIPS↓ | MUSIQ↑ | MANIQA↑ | TOPIQ↑ | LIQE↑ |
| FFTFormer (Kong et al., 2023) | 24.35 | 0.5867 | 0.4463 | 23.83 | 0.5713 | 0.4751 | 30.1430 | 0.4517 | 0.2655 | 1.1819 |
| +ORNet (**Ours**) | **24.41** | **0.6179** | **0.4070** | **23.97** | **0.6057** | **0.4233** | **41.8760** | **0.4699** | **0.3188** | **1.4321** |

## 5.2 RESULTS

### 5.2.1 IN-DISTRIBUTION QUANTITATIVE RESULTS

Table 1 and Table 2 report the quantitative results on the GoPro deblurring and SIDD denoising datasets, respectively. As ORNet is model agnostic, we apply it on top of representative base models from two restoration tasks. For image deblurring, we integrate ORNet into AdaRevD (Mao et al., 2024) and FFTformer (Kong et al., 2023); for image denoising, we use NAFNet (Chen et al., 2022) and Xformer (Zhang et al., 2024). We compare against diverse state-of-the-art methods, including Restormer (Zamir et al., 2022), ResShift (Yue et al., 2024), IR-SDE (Luo et al., 2023), DiffIR (Xia et al., 2023), HIDiff (Chen et al., 2024b) for deblurring, and AP-BSN (Lee et al., 2022), MIRNet-v2 (Zamir et al., 2020), Restormer for denoising.

Models trained with $\ell_1$ or $\ell_2$-loss and diffusion-based models exhibit comparable performance in terms of both no-reference and VLM-based scores. This suggests that the restoration quality of existing models is upper-bounded by the quality of the original ground truth. In contrast, our method leverages an enhanced ground truth, thereby achieving a significant improvement in perceptual quality. We note that reference-based metrics are invalid for evaluation, as there are two different ground truths: the original and the enhanced.

### 5.2.2 OUT-OF-DISTRIBUTION QUANTITATIVE RESULTS

To evaluate the generalization performance of our refinement network, we conduct experiments in out-of-distribution (OOD) settings. These are constructed by augmenting the inputs of the GoPro test set (Nah et al., 2017) with additional, unseen degradations: one set with Gaussian blur and another with white noise. We posit that existing state-of-the-art deblurring model FFTformer (Kong et al., 2023) overfit to the specific degradation characteristics of their GoPro training data, causing their performance to degrade sharply in such OOD conditions. In contrast, our ORNet is not trained for a specific degradation; it robustly enhances the output of any given restoration model. As demonstrated in Tables 3 and 4, applying ORNet leads to a substantial increase in perceptual quality. Simultaneously, reference-based metrics improve against both the original GT and our enhanced GT, which validates that ORNet also effectively preserves semantic details. Additional analysis of the generalization performance of our ORNet is provided in Appendix B.3.

### 5.2.3 QUALITATIVE RESULTS

We present qualitative results highlighting two key aspects of our approach: (1) the enhanced ground truth images generated by our supervision framework, and (2) the restoration outputs refined by ORNet. As shown in Figure 3, our enhanced GT preserves the semantic content of the original GT while providing sharper and more perceptually pleasing details. In Figures 4 and 5, we compare restored outputs on the GoPro and SIDD datasets. When combined with existing restoration models, ORNet consistently improves perceptual quality in fine details such as the cracks between stones in Figure 4 and the sharpness of text and edges in Figure 5, yielding outputs with sharper and cleaner details than those in the original GTs.

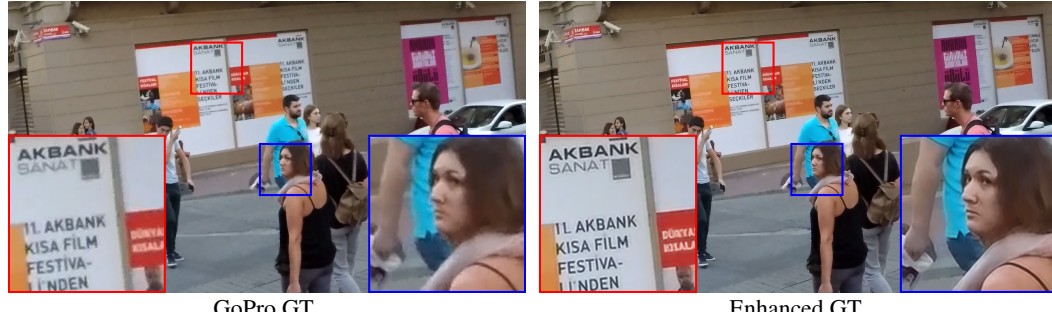

Figure 3: Visualization of enhanced ground truth. Our enhanced GT not only exhibits sharper text and superior perceptual quality but also maintains semantic consistency. Zoom in for better visualizaiton.

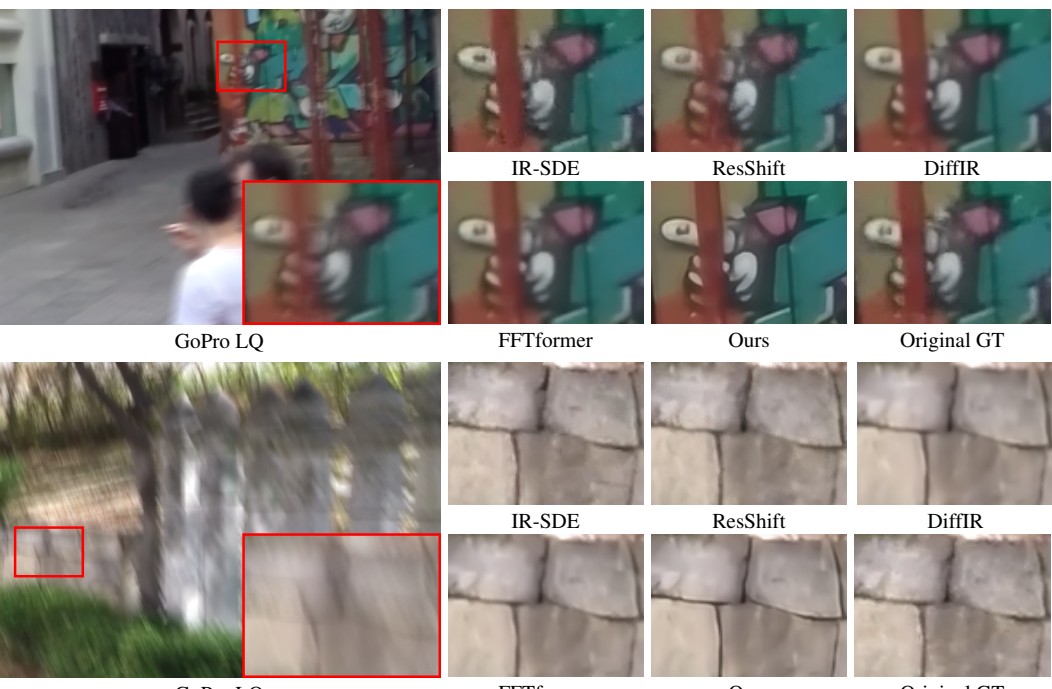

Figure 4: Qualitative comparison of state-of-the-art deblurring methods, including ours (ORNet applied to FFTformer), on the GoPro dataset. Our method significantly improves the visual quality of the deblurred image. Zoom in for better visualization.

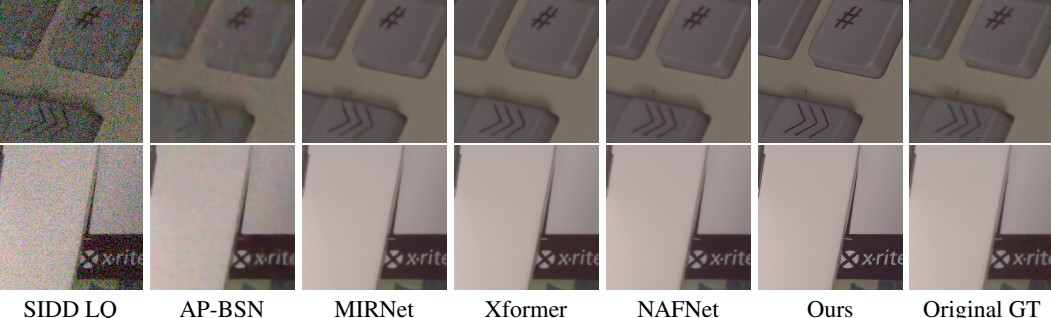

Figure 5: Qualitative comparison of state-of-the-art denoising methods, including ours (ORNet applied to NAFNet), on the SIDD dataset. Our method significantly improves the visual quality of the denoised image. Zoom in for better visualization.

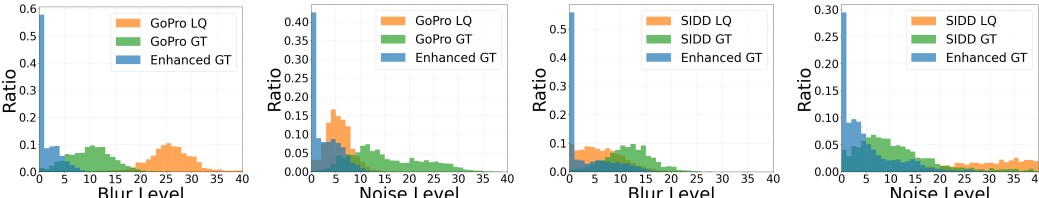

Figure 6: Our enhanced GT images demonstrate significantly improved blur and noise levels, assessed with KonIQ++ (Su et al., 2021). Histograms compare low quality (LQ), ground truth (GT), and enhanced GT images from GoPro (Nah et al., 2017) and SIDD (Abdelhamed et al., 2018) datasets.

### 5.3 ANALYSIS

**KonIQ++ analysis** Figure 6 shows the results of our supervision enhancement framework. We assess the quality of images using KonIQ++ (Su et al., 2021) blur level ($\downarrow$) and noise level ($\downarrow$). We observe that the ground truth (GT) images in the GoPro (Nah et al., 2017) dataset, captured using single high-shutter-speed frames from action cameras, tend to be relatively noisy. In addition, the GT images in the SIDD (Abdelhamed et al., 2018) dataset exhibit high blur scores, indicating that the averaging process used to obtain GT images introduces blurriness. Our enhancement framework effectively improves the quality of such suboptimal GT images, reducing both blur and noise.

**User Study** We conduct two user studies to evaluate the perceptual quality of our supervision enhancement and output refinement network. Both studies involve 70 participants, each presented with 25 randomly sampled images from the GoPro dataset. For supervision enhancement, participants are asked to compare the original ground truth (baseline) and enhanced ground truth (ours), based on how well each image appears to restore the low-quality input. For the output refinement network, participants evaluate which output, FFTformer (baseline) or FFTformer + ORNet (ours), provides a better restoration of the low-quality input. As shown in Figure 7, both our enhanced ground truth images and refinement outputs received significantly higher preference scores.

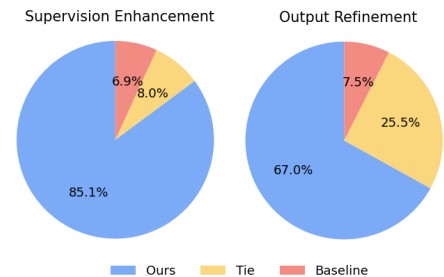

Figure 7: User study results. Participants consistently preferred our enhanced GT and ORNet outputs to the baselines.

**Efficiency Comparisons** Table 5 summarizes the number of parameters and multiply-accumulate operations (MACs) of our refinement network and existing restoration networks. The numbers are calculated with an input of $512{\times}512$. Our refinement network (ORNet) is significantly lightweight compared to the base restoration models. This efficiency allows our method to be easily integrated into existing architectures without incurring a computational overhead.

Table 5: Parameters and MACs of the restoration networks and output refinement network.

| Architecture | Params. (M) | MACs (G) |
|---|---|---|
| AdaRev | 68.0 | 1386 |
| FFTformer | 14.9 | 525 |
| NAFNet | 115.9 | 254 |
| Xformer | 25.1 | 571 |
| ORNet **(Ours)** | 4.5 | 20 |

## 6 CONCLUSION

We introduce a novel supervision enhancement framework that addresses the critical limitation of suboptimal ground truth images in real-world image restoration. By generating perceptually superior GT variants via super-resolution and optimally fusing them with original GTs in the frequency domain using adaptive masks, we achieve enhanced supervision targets. Comprehensive evaluations, including user studies and diverse metrics, confirm that our method successfully balances fidelity with significantly improved perceptual quality. This enhanced supervision then enables the training of a lightweight, model-agnostic refinement network, which can seamlessly integrate with existing restoration models to further boost their output. We emphasize that our framework offers a practical path toward higher-fidelity and more visually compelling results in real-world restoration scenarios.

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

# Appendix

# Beyond the Ground Truth:
# Enhanced Supervision for Image Restoration

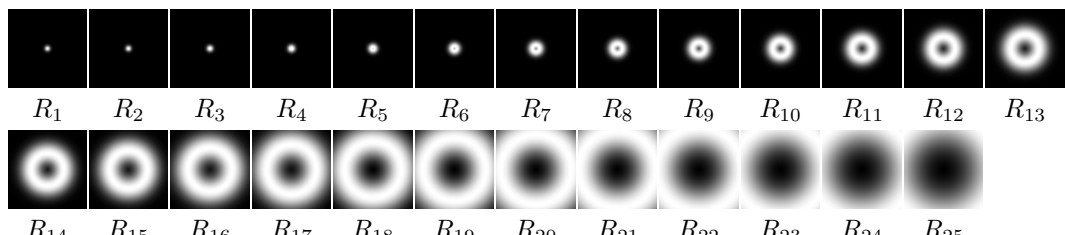

Figure S.1: Visualization of the predefined masks $R_1$-$R_{25}$. It demonstrates denser partitioning in the low-frequency domain and broader partitioning in the high-frequency domain.

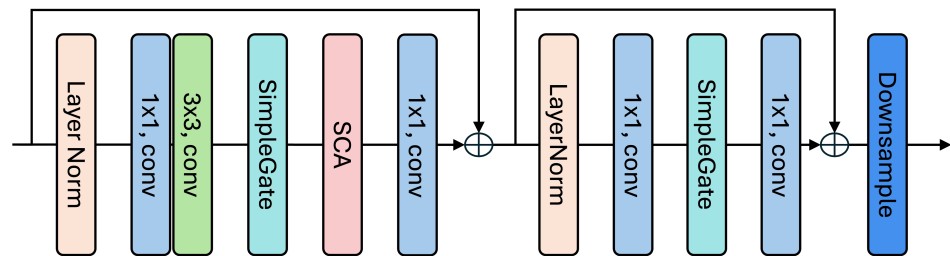

Figure S.2: The details of the NAFBlock.

## A ADDITIONAL IMPLEMENTATION DETAILS

### A.1 BASIS MASKS

In our main paper, Equation (2) defines the $b$-th ring-shaped Gaussian basis mask $R_b$ using parameters $\mu_b$ and $\sigma_b$. Here, $\mu_b$ represents the radial distance from the frequency-domain center where the mask has its peak, and $\sigma_b$ indicates the spread of the mask. We construct a total of $B = 25$ Gaussian basis masks. The first mask is centered at $\mu_1 = 0$ with a standard deviation of $\sigma_1 = 0.05$. For $b = 1, \ldots, B$, the peak positions $\mu_b$ are arranged by quadratically spacing values between 0 and $\sqrt{H^2 + W^2}/2$, where $H$ and $W$ are height and width of the image, yielding denser coverage near the DC component and sparser placement at higher frequencies. Simultaneously, the spreads $\sigma_b$ increase quadratically from $0.05$ up to $0.55$, providing narrower rings at low frequencies and broader ones at high frequencies. This design ensures fine control around the low-frequency region and efficient coverage of the full frequency range. The all predefined masks are visualized in Figure S.1.

### A.2 ARCHITECTURE DETAILS

Figure S.2 shows the details of the NAFBlock, utilized within the frequency mask generator illustrated in Figure 2 (b). The foundational block structures, including the Simple Gate and Simplified Channel Attention (SCA), are adopted from the NAFNet architecture (Chen et al., 2022). An additional Downsample operation, composed of a convolution with a kernel size of $2 \times 2$ and a stride of 2, is incorporated into this NAFBlock variant. And the proposed output refinement network (ORNet) consists of the 4 encoder, 1 middle, and 4 decoder blocks, employing the NAFBlock as their fundamental building unit. The encoder block is same with figure S.2. The middle block does not incorporate last downsampling operation. The decoder block implements an upsampling instead of downsampling: it first doubles the channel dimensionality using a $1 \times 1$ convolution, followed by a pixel shuffle module that doubles both the height and width of the feature maps, following the NAFNet architecture (Chen et al., 2022).

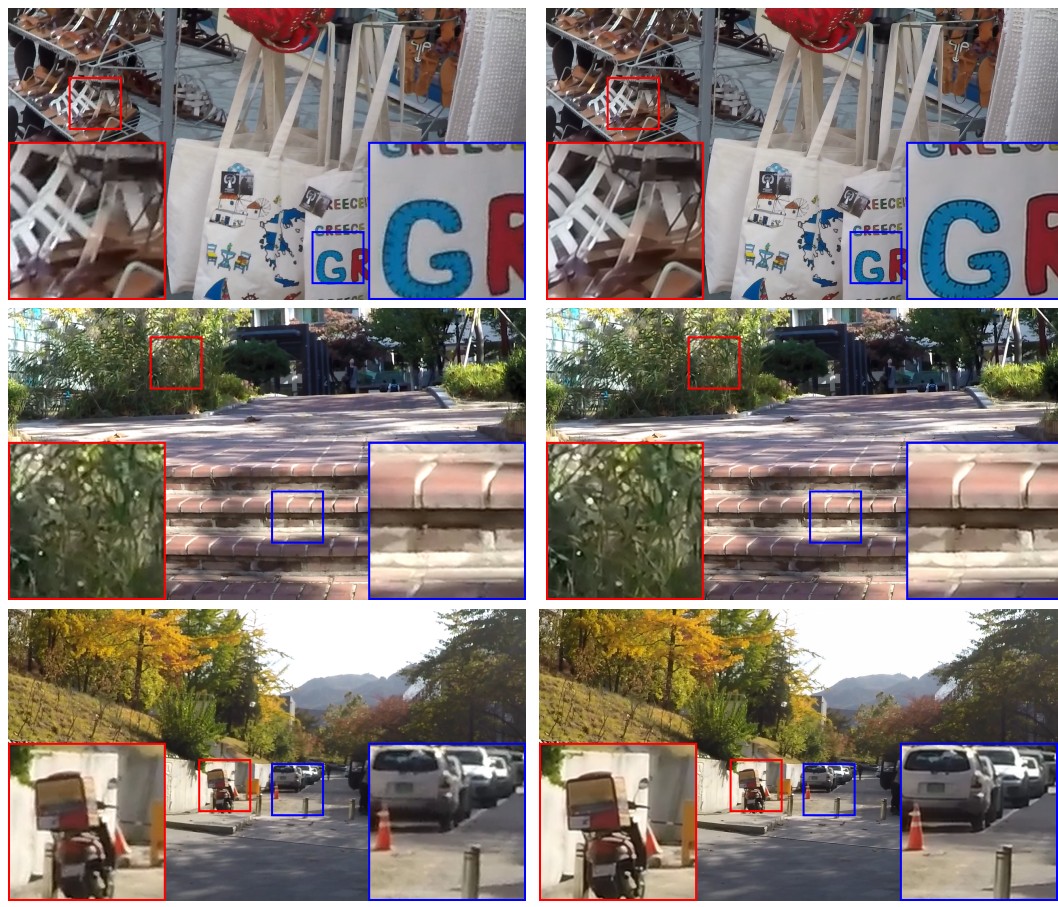

GoPro GT          Enhanced GT

Figure S.3: Visualization of enhanced ground truth. Our enhanced GT not only exhibits sharper text and superior perceptual quality but also maintains semantic consistency. Zoom in for better visualizaiton.

## B  ADDITIONAL EXPERIMENTS

### B.1  ENAHNCED GT VISUALIZATION

In Figure S.3, we present our additional qualitative results of generated enhanced GTs. Our results clearly demonstrate superior perceptual quality by effectively removing remaining degradation such as noise and blur from the original GTs, while maintaining semantic consistency. In Figure S.4, we compare our supervision enhancement with a simple super-resolved variant. Whereas the super-resolved variant primarily sharpens details and brightens colors, our method delivers richer perceptual improvements while preserving both semantic structure and overall color tone.

### B.2  ABLATION STUDY ON ENHANCEMENT LEVEL $\lambda$

**Quantative result**  Tables S.1, S.2, S.3 and S.4 show the full expanded versions of Tables 1–4 in the main paper, with different $\lambda$ values (0.1, 0.3, 0.5, 0.7, and 0.9). A consistent trend observed is the steady improvement in perceptual quality metric scores as the value of $\lambda$ increases. However, as shown in Tables S.1 and S.2, the VLM-based score exhibits a different behavior depending the dataset. This suggests that for datasets with very low initial quality, such as SIDD, a larger $\lambda$ leads to continuous improvement. Conversely, for datasets with relatively higher quality, like GoPro, excessive enhancement may introduce perceptually adverse artifacts, such as over-saturation and over-sharpening. Furthermore, in the out-of-distribution (OOD) settings shown in Tables S.3 and S.4, we observe that an excessively high $\lambda$ value can degrade reference-based performance against both the original and enhanced GTs. This appears to be because when $\lambda$ is too large, the ORNet applies

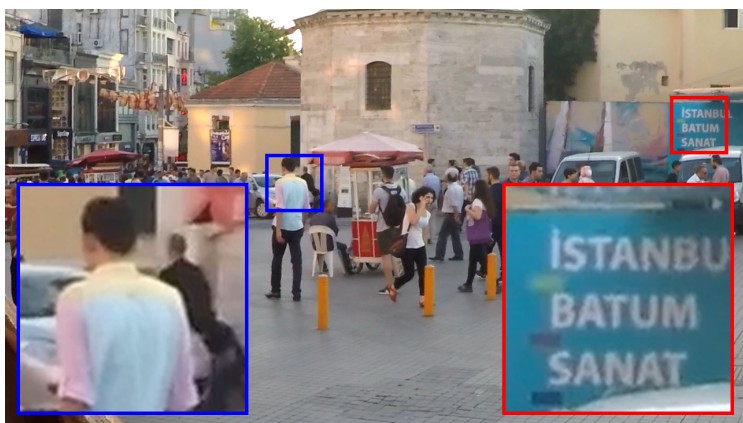

Original GT

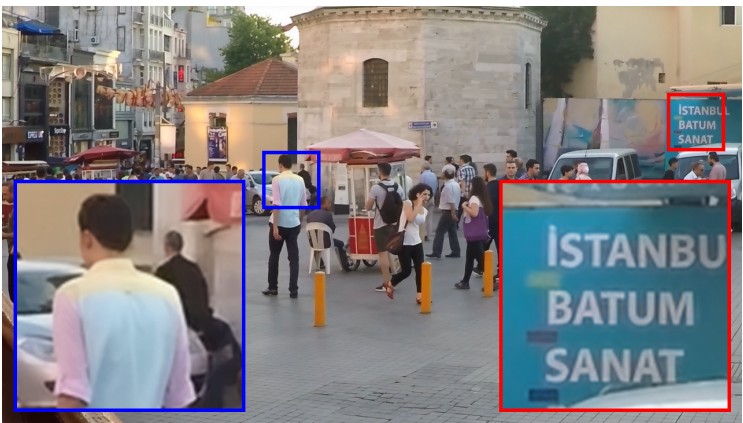

Enhanced GT (Ours)

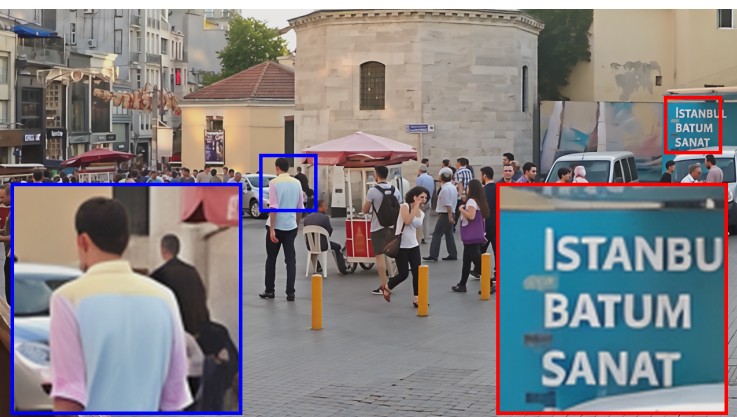

SR Variant (x2)

Figure S.4: Comparison of our supervision enhancement with SR (x2) variant. Zoom in for better visualization of the semantic details and overall color tone.

changes in color tone and further enhancements that go beyond removing the remaining degradations (blur, noise), resulting in a reference based performance drop. This highlights the importance of selecting an appropriate $\lambda$ to achieve a balance between enhancement and fidelity.

**User study** We conducted a user study to validate the perceptual quality of the output images of ORNet with different $\lambda$ values. Following the same protocol as described in the main paper, we

Table S.1: Full expanded table of Table 1 in the main paper with diverse enhancement levels. The results are evaluated on the GoPro dataset (Nah et al., 2017) with the settings in the main paper.

| Method | Perceptual Quality Metrics | | | | VLM-based Metrics | |
|---|---|---|---|---|---|---|
| | MUSIQ↑ | MANIQA↑ | TOPIQ↑ | LIQE↑ | VisualQuality-R1↑ | Q-Insight↑ |
| AdaRevD (Mao et al., 2024) | 45.49 | 0.5363 | 0.3393 | 1.5656 | 4.1737 | 3.4386 |
| + ORNet ($\lambda$=0.1) | 47.42 | 0.5451 | 0.3490 | 1.6256 | 4.1939 | 3.4605 |
| + ORNet ($\lambda$=0.3) | 64.25 | 0.5916 | 0.4880 | 2.4291 | 4.1952 | 3.5206 |
| + ORNet ($\lambda$=0.5) | 69.11 | 0.6161 | 0.5893 | 2.9711 | 4.0686 | 3.5163 |
| + ORNet ($\lambda$=0.7) | 69.72 | 0.6330 | 0.6098 | 3.2308 | 4.0025 | 3.5063 |
| + ORNet ($\lambda$=0.9) | 69.73 | 0.6420 | 0.6195 | 3.3870 | 3.9146 | 3.4838 |
| FFTformer (Kong et al., 2023) | 46.47 | 0.5420 | 0.3456 | 1.6131 | 4.0942 | 3.4569 |
| + ORNet ($\lambda$=0.1) | 48.07 | 0.5484 | 0.3537 | 1.6619 | 4.2334 | 3.4764 |
| + ORNet ($\lambda$=0.3) | 64.57 | 0.5949 | 0.4924 | 2.4664 | 4.1995 | 3.5278 |
| + ORNet ($\lambda$=0.5) | 69.18 | 0.6189 | 0.5905 | 2.9944 | 4.0918 | 3.5234 |
| + ORNet ($\lambda$=0.7) | 69.76 | 0.6352 | 0.6104 | 3.2444 | 4.0262 | 3.5162 |
| + ORNet ($\lambda$=0.9) | 69.76 | 0.6440 | 0.6198 | 3.3953 | 3.9600 | 3.4957 |

Table S.2: Full expanded table of Table 2 in the main paper with diverse enhancement levels. The results are evaluated on the SIDD dataset (Abdelhamed et al., 2018) with the same settings as in the main paper.

| Method | Perceptual Quality Metrics | | | | VLM-based Metrics | |
|---|---|---|---|---|---|---|
| | MUSIQ↑ | MANIQA↑ | TOPIQ↑ | LIQE↑ | VisualQuality-R1↑ | Q-Insight↑ |
| Xformer (Zhang et al., 2024) | 22.57 | 0.3828 | 0.2472 | 1.2040 | 1.0759 | 1.6710 |
| + ORNet ($\lambda$=0.1) | 26.23 | 0.3819 | 0.2738 | 1.3238 | 1.1029 | 1.7972 |
| + ORNet ($\lambda$=0.3) | 35.68 | 0.4310 | 0.3710 | 1.9510 | 1.3228 | 2.1227 |
| + ORNet ($\lambda$=0.5) | 37.53 | 0.4517 | 0.3908 | 2.1195 | 1.3835 | 2.1827 |
| + ORNet ($\lambda$=0.7) | 37.99 | 0.4615 | 0.3968 | 2.1711 | 1.3975 | 2.2157 |
| + ORNet ($\lambda$=0.9) | 38.05 | 0.4661 | 0.3989 | 2.1867 | 1.4262 | 2.2176 |
| NAFNet (Chen et al., 2022) | 22.73 | 0.3937 | 0.2458 | 1.2189 | 1.0826 | 1.7060 |
| + ORNet ($\lambda$=0.1) | 26.39 | 0.3917 | 0.2776 | 1.3228 | 1.1224 | 1.8217 |
| + ORNet ($\lambda$=0.3) | 35.87 | 0.4380 | 0.3776 | 1.9591 | 1.3513 | 2.1584 |
| + ORNet ($\lambda$=0.5) | 37.89 | 0.4605 | 0.3977 | 2.1394 | 1.4030 | 2.2269 |
| + ORNet ($\lambda$=0.7) | 38.40 | 0.4709 | 0.4039 | 2.1934 | 1.4321 | 2.2498 |
| + ORNet ($\lambda$=0.9) | 38.46 | 0.4758 | 0.4057 | 2.2088 | 1.4492 | 2.2501 |

Table S.3: Evaluation on an OOD environment of the GoPro test set, where an additional Gaussian blur ($\sigma = 2.5$) is applied to the blurry input images.

| Method | Original GT | | | Enhanced GT | | | Perceptual Quality Metrics | | | |
|---|---|---|---|---|---|---|---|---|---|---|
| | PSNR↑ | SSIM↑ | LPIPS↓ | PSNR↑ | SSIM↑ | LPIPS↓ | MUSIQ↑ | MANIQA↑ | TOPIQ↑ | LIQE↑ |
| FFTFormer | 24.5688 | 0.7532 | 0.4714 | 23.6891 | 0.7224 | 0.5441 | 22.3812 | 0.2284 | 0.1832 | 1.0108 |
| +ORNet ($\lambda$=0.1) | 24.5898 | 0.7550 | 0.4592 | 23.7141 | 0.7245 | 0.5310 | 23.1592 | 0.2481 | 0.1843 | 1.0111 |
| +ORNet ($\lambda$=0.3) | 24.5774 | 0.7670 | 0.3429 | 23.8124 | 0.7405 | 0.3777 | 42.9131 | 0.2638 | 0.2646 | 1.0656 |
| +ORNet ($\lambda$=0.5) | 24.5789 | 0.7742 | 0.3107 | 23.8835 | 0.7500 | 0.3334 | 49.6099 | 0.2689 | 0.3324 | 1.2576 |
| +ORNet ($\lambda$=0.7) | 24.4442 | 0.7759 | 0.3042 | 23.8007 | 0.7524 | 0.3253 | 50.9512 | 0.2950 | 0.3487 | 1.3977 |
| +ORNet ($\lambda$=0.9) | 24.2407 | 0.7754 | 0.3036 | 23.6454 | 0.7524 | 0.3243 | 51.5943 | 0.3174 | 0.3174 | 1.0656 |

evaluated three levels of the refinement weight $\lambda \in \{0.1, 0.3, 0.5\}$. As shown in Table S.5, the results indicate that while both $\lambda = 0.3$ and $\lambda = 0.5$ achieved similarly high preference rates, $\lambda = 0.3$ yielded the lowest loss rate.

**Qualitative results** Figure S.5 visualizes the results of our Ground Truth (GT) enhancement with varying values of the hyperparameter $\lambda$. As $\lambda$ increases, the perceptual quality may be enhanced, but this can introduce undesirable artifacts such as altered color tones and semantic changes that deviate from the original GT. In contrast, an optimally chosen $\lambda$ effectively removes residual noise and blur,

Table S.4: Evaluation on an OOD environment of the GoPro test set, where an additional white noise ($\sigma = 9$) is applied to the blurry input images.

| Method | Original GT | | | Enhanced GT | | | Perceptual Quality Metrics | | | |
|---|---|---|---|---|---|---|---|---|---|---|
| | PSNR↑ | SSIM↑ | LPIPS↓ | PSNR↑ | SSIM↑ | LPIPS↓ | MUSIQ↑ | MANIQA↑ | TOPIQ↑ | LIQE↑ |
| FFTFormer | 24.3574 | 0.5867 | 0.4463 | 23.8352 | 0.5713 | 0.4751 | 30.1430 | 0.4517 | 0.2655 | 1.1819 |
| +ORNet ($\lambda$=0.1) | 24.3804 | 0.5886 | 0.4438 | 23.8616 | 0.5734 | 0.4720 | 30.5873 | 0.4528 | 0.2664 | 1.1909 |
| +ORNet ($\lambda$=0.3) | 24.4088 | 0.6179 | 0.4070 | 23.9693 | 0.6057 | 0.4233 | 41.8760 | 0.4699 | 0.3188 | 1.4321 |
| +ORNet ($\lambda$=0.5) | 24.4819 | 0.6771 | 0.3549 | 24.1483 | 0.6671 | 0.3602 | 55.2554 | 0.5204 | 0.4034 | 1.8986 |
| +ORNet ($\lambda$=0.7) | 24.3252 | 0.6943 | 0.3372 | 24.0463 | 0.6852 | 0.3397 | 58.9384 | 0.5473 | 0.4212 | 2.1779 |
| +ORNet ($\lambda$=0.9) | 24.0224 | 0.7018 | 0.3313 | 23.7722 | 0.6928 | 0.3335 | 60.1486 | 0.5618 | 0.4295 | 2.2923 |

Table S.5: User study with various $\lambda$ values. Participants consistently preferred our ORNet outputs to the baselines.

| $\lambda$ | Win (%) | Tie (%) | Lose (%) |
|---|---|---|---|
| 0.1 | 25.7 | 56.1 | 18.1 |
| 0.3 | 67.0 | 25.5 | 7.5 |
| 0.5 | 68.5 | 21.4 | 10.1 |

leading to a perceptually improved image while preserving the color and semantic integrity of the original.

### B.3 GENERALIZATION ON OUT-OF-DISTRIBUTION DATASET AND UNSEEN TASK

**Quantitative results**   Furthermore, to evaluate generalization performance, we test our method on HIDE (Shen et al., 2019) as an out-of-distribution (OOD) deblurring benchmark. We additionally evaluate it on LOL (Wei et al., 2018), a low-light enhancement benchmark, as an unseen restoration task. For evaluating low light-enhancement, we adapt our refinement network to Retinexformer (Cai et al., 2023) and CIDNet (Yan et al., 2025). As shown in Table S.6 and Table S.7, our approach consistently enhances perceptual quality even for the unknown dataset and task, demonstrating its strong generalization capabilities.

**Qualitative results**   Figure S.6 presents the qualitative results of our ORNet ($\lambda$=0.3) when applied to the output of CIDNet (Yan et al., 2025) on the LOL low light enhancement dataset (Wei et al., 2018). Our ORNet effectively enhances the overall quality of the output, resulting in a more visually appealing image.

### B.4 FINE-TUNING RESTORATION MODEL WITH ENHANCED SUPERVISION

Our enhanced supervision can be utilized in various ways, including directly finetuning existing restoration models. Table S.8 presents these results. The first row, FFTformer, shows the performance of a model pretrained on the original GoPro dataset. The subsequent row illustrates the results when our modular refinement network is applied to its output. FFTformer* indicates the results after finetuning the pretrained FFTformer with our enhanced supervision. All presented results are obtained with $\lambda = 0.3$. We observe that finetuning with our enhanced supervision allows existing restoration models to achieve high perceptual quality in terms of no-reference metrics, significantly improving the overall scores. However, this approach has a limitation: each network must be retrained with its corresponding enhanced supervision. In contrast, our modular refinement network can be attached to various restoration models as a single, unified module, achieving better scores than direct finetuning.

### B.5 ADDITIONAL ABLATION STUDY ON MASK DESIGN

Figure S.7 presents a comparative analysis of frequency masks generated by three distinct methods: our conditional frequency mask generator with ring-shaped Gaussian basis, and an element-wise baseline in frequency domain and spatial domain. Our method generates frequency masks as a

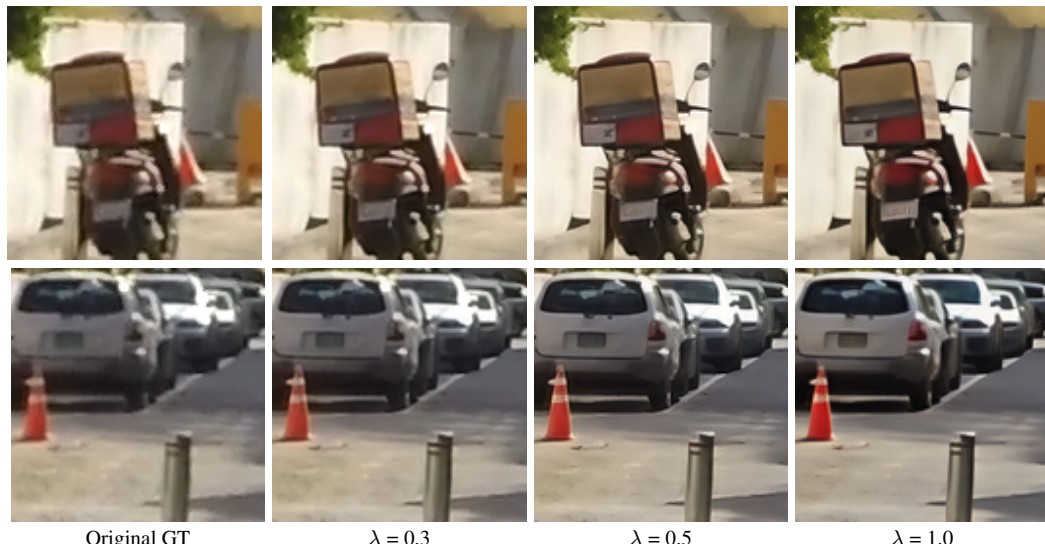

| Original GT | $\lambda = 0.3$ | $\lambda = 0.5$ | $\lambda = 1.0$ |

Figure S.5: Qualitative comparison of GT enhancement with varying $\lambda$ values. Excessively large $\lambda$ values increase the risk of hallucinations, such as color shifts and semantic deviations from the original GT. Zoom in for better visualization.

Table S.6: The results are evaluated on the HIDE dataset (Shen et al., 2019) with the same settings as in the main paper.

| Method | No Ref. | | | | VLM-based. | |
| | MUSIQ↑ | MANIQA↑ | TOPIQ↑ | LIQE↑ | VisualQuality-R1↑ | Q-Insight↑ |
|---|---|---|---|---|---|---|
| AdaRevD (Mao et al., 2024) | 55.91 | 0.5882 | 0.4064 | 2.0823 | 4.3013 | 3.4613 |
| + ORNet ($\lambda$=0.1) | 58.34 | 0.5967 | 0.4211 | 2.1717 | 4.3630 | 3.5226 |
| + ORNet ($\lambda$=0.3) | 68.48 | 0.6282 | 0.5375 | 2.6209 | 4.3341 | 3.5697 |
| + ORNet ($\lambda$=0.5) | 71.70 | 0.6443 | 0.6314 | 2.9715 | 4.2557 | 3.5661 |
| + ORNet ($\lambda$=0.7) | 72.03 | 0.6549 | 0.6409 | 3.1910 | 4.1948 | 3.5494 |
| + ORNet ($\lambda$=0.9) | 71.57 | 0.6493 | 0.6361 | 3.2012 | 4.1217 | 3.5372 |
| FFTFormer (Kong et al., 2023) | 54.42 | 0.5768 | 0.3978 | 2.0440 | 4.3176 | 3.4733 |
| + ORNet ($\lambda$=0.1) | 57.72 | 0.5943 | 0.4125 | 2.1316 | 4.3599 | 3.5229 |
| + ORNet ($\lambda$=0.3) | 68.19 | 0.6256 | 0.5295 | 2.5832 | 4.3351 | 3.5691 |
| + ORNet ($\lambda$=0.5) | 71.42 | 0.6416 | 0.6237 | 2.9388 | 4.2458 | 3.5645 |
| + ORNet ($\lambda$=0.7) | 71.67 | 0.6524 | 0.6329 | 3.1664 | 4.1871 | 3.5476 |
| + ORNet ($\lambda$=0.9) | 71.08 | 0.6461 | 0.6278 | 3.1784 | 4.1332 | 3.5341 |

weighted combination of smooth Gaussian basis, promoting spatial coherence. The base mask $M_0$ is applied to the original ground truth and preserves low-frequency content, while the additional masks $M_1$, $M_2$, and $M_3$ selectively incorporate high-frequency components from super-resolved variants. This structured decomposition enables fine-grained and interpretable frequency control across spatial regions.

In contrast, the element-wise baseline in frequency domain uses a simple U-Net to directly predict mask values for each spatial cordinate in frequency domain. Although it exhibits a similar frequency selection tendency, the resulting masks are spatially inconsistent, often leading to artifacts in the final enhanced ground truth. Additionally, the element-wise baseline in spatial domain utilizes the same network designed above with the only modification being that the mixup is performed in the spatial domain instead of the frequency domain. As shown in Figure S.7, employing element–wise mixup directly in the frequency and spatial domain leads to unstable mask generation.

These artifacts arise in part from the use of perceptual loss guided by no-reference image quality assessment (IQA) models, which struggle to detect subtle unnatural distortions. As a result, relying solely on IQA-based perceptual loss makes it difficult to avoid such inconsistency-induced artifacts.

Table S.7: The results are evaluated on the LOL dataset (Wei et al., 2018) with the same settings as in the main paper.

| Method | No Ref. | | | | VLM-based. | |
| | MUSIQ↑ | MANIQA↑ | TOPIQ↑ | LIQE↑ | VisualQuality-R1↑ | Q-Insight↑ |
|---|---|---|---|---|---|---|
| Retinexformer (Cai et al., 2023) | 63.15 | 0.5870 | 0.5419 | 2.8354 | 3.4400 | 3.3153 |
| + ORNet ($\lambda$=0.1) | 64.28 | 0.5944 | 0.5488 | 2.9363 | 3.5200 | 3.4080 |
| + ORNet ($\lambda$=0.3) | 72.80 | 0.6652 | 0.6435 | 3.9872 | 3.5200 | 3.5047 |
| + ORNet ($\lambda$=0.5) | 74.76 | 0.7043 | 0.6669 | 4.4898 | 3.7667 | 3.6507 |
| + ORNet ($\lambda$=0.7) | 75.18 | 0.7153 | 0.6715 | 4.5909 | 3.8467 | 3.7107 |
| + ORNet ($\lambda$=0.9) | 75.31 | 0.7138 | 0.6735 | 4.6085 | 3.7400 | 3.7067 |
| CIDNet (Yan et al., 2025) | 69.51 | 0.6256 | 0.6288 | 3.8336 | 3.9133 | 3.6967 |
| + ORNet ($\lambda$=0.1) | 70.45 | 0.6329 | 0.6366 | 3.9853 | 4.0400 | 3.7447 |
| + ORNet ($\lambda$=0.3) | 74.88 | 0.7035 | 0.7108 | 4.7373 | 3.9000 | 3.7613 |
| + ORNet ($\lambda$=0.5) | 75.77 | 0.7295 | 0.7191 | 4.8546 | 4.0000 | 3.7667 |
| + ORNet ($\lambda$=0.7) | 75.97 | 0.7350 | 0.7183 | 4.8512 | 4.0467 | 3.7760 |
| + ORNet ($\lambda$=0.9) | 76.03 | 0.7324 | 0.7176 | 4.8431 | 4.0267 | 3.7447 |

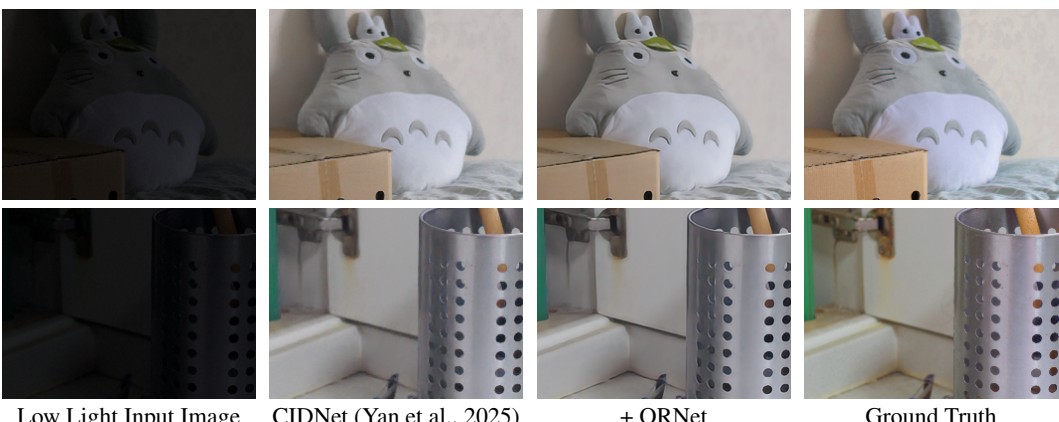

Low Light Input Image     CIDNet (Yan et al., 2025)     + ORNet     Ground Truth

Figure S.6: Qualitative results of our ORNet when applied to the output of CIDNet (Yan et al., 2025) on the LOL low light enhancement dataset (Wei et al., 2018). Zoom in for better visualization.

Our Gaussian basis constraint allows stable control over frequency content, enabling effective use of perceptual loss while suppressing artifacts caused by spatial irregularities.

### B.6 ABLATION STUDY ON GROUND TRUTH VARIANTS

To generate an enhanced ground truth, we first employ a super-resolution model to create ground truth variants. For this purpose, we utilize upscale factors of 2, 3, and 4. By applying a frequency mixup strategy to these diverse ground truth variants, we successfully construct an enhanced ground truth. As an ablation study, we conduct an experiment with different grount truth variant setting, solely using upscale factor of 4. Using these 4x variants, we follow our supervision enhancement framework and output refinement network. The results are presented in Table S.9. As shown, ORNet_4x, representing the model trained exclusively with the 4 upscale factor, achieved final scores that are marginally lower than those obtained by the model trained with a mixture of variants. This indicates that the incorporation of diverse GT variants is beneficial for achieving optimal final enhancement.

### B.7 ABLATION ON USING ISR NETWORK INSTEAD OF ORNET

Our output refinement network (ORNet) is trained to refine the output of any existing image restoration model, which is trained with enhanced supervision. To enhance the output of the restoration model, we could also directly apply the diffusion based image super-resolution (ISR) model used in our framework. However, ISR model, which is designed to enhance the perceptual quality, often

Table S.8: Comparison with our output refinement network (ORNet) and directly finetuning the restoration model with our enhanced supervision. FFTformer* denotes a model finetuned on the GoPro training dataset where the original ground truth was replaced with an enhanced ground truth generated with $\lambda = 0.3$, using a pretrained model trained on the original GoPro dataset.

| Method | No Ref. | | | |
| | MUSIQ↑ | MANIQA↑ | TOPIQ↑ | LIQE↑ |
| --- | --- | --- | --- | --- |
| FFTformer (Kong et al., 2023) | 46.47 | 0.5420 | 0.3456 | 1.6130 |
| + ORNet | 64.57 | 0.5949 | 0.4924 | 2.4664 |
| FFTformer* (Kong et al., 2023) | 60.54 | 0.5854 | 0.4509 | 2.2359 |

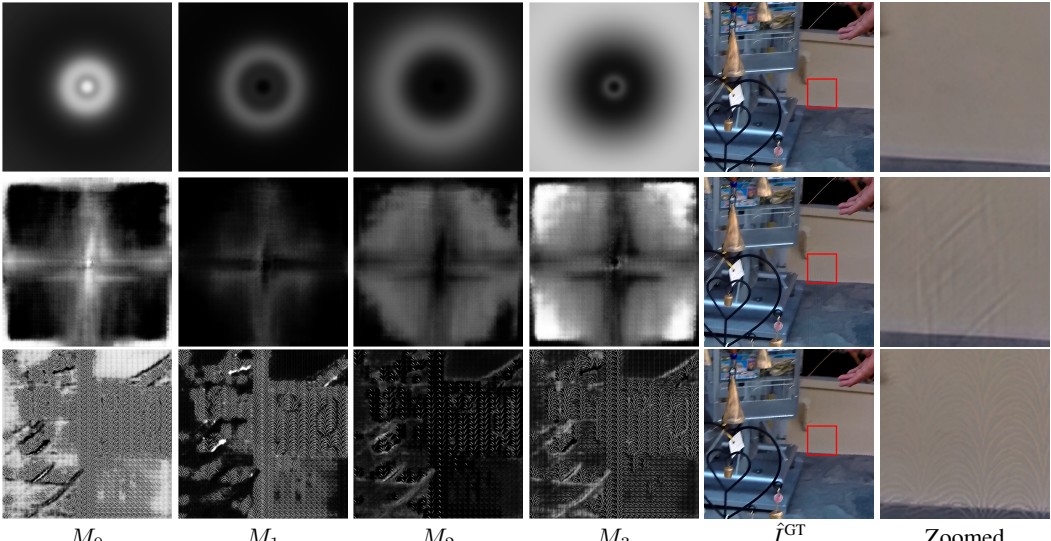

$M_0$ $\qquad$ $M_1$ $\qquad$ $M_2$ $\qquad$ $M_3$ $\qquad$ $\hat{I}^{GT}$ $\qquad$ Zoomed

Figure S.7: The top row shows the results when the conditional frequency mask generator is trained using our method. The second row shows the results when it is trained in an element-wise manner without ring-shaped Gaussian basis in frequency domain. The bottom row shows the results when it is trained in an element-wise manner in spatial domain. $M_i$ denotes the generated frequency masks, and $\hat{I}^{GT}$ represents the enhanced ground truth generated using these masks. Zoom in for better visualization.

hallucinates details that are not present in the input image when applied directly. Figure S.8 shows the results of applying the ISR model directly to the output of the restoration model, where the refined output destroys the textual detail.

## B.8 MASK VISUALIZATION ON DIVERSE ENHANCEMENT LEVEL $\lambda$

Figure S.7 shows the visualization of the generated masks with different $\lambda$ values. When $\lambda$ is small, the generated masks are mostly focused on $M_0$, dedicated for original ground truth image. As $\lambda$ increases, generated masks cover a diverse other ground truth variants.

## B.9 TRAINING STABILITY

To assess the training stability of our modular output refinement network (ORNet), we trained the ORNet with five independent times using distinct random seeds. Following training, each refinement network was applied to the outputs of a pretrained AdaRevD (Mao et al., 2024) on the GoPro test dataset. Then, the standard deviation is calculated for each metric with $\lambda = 0.3$. The resulting standard deviations were as follows: PSNR (0.023), SSIM (0.0002), LPIPS (0.0005), DISTS (0.0005), MUSIQ (0.097), MANIQA (0.0007), TOPIQ (0.001), and LIQE (0.01). The observed standard deviations for each metric are notably low. This outcome indicates a high degree of stability in our training procedure for the output refinement network.

Table S.9: Comparison of our ORNet with only using ground truth variant with upscale factor 4.

| Method | MUSIQ↑ | MANIQA↑ | TOPIQ↑ | LIQE↑ |
|---|---|---|---|---|
| FFTformer (Kong et al., 2023) | 46.47 | 0.5420 | 0.3456 | 1.6131 |
| + ORNet ($\lambda$=0.1) | 48.07 | 0.5484 | 0.3537 | 1.6619 |
| + ORNet ($\lambda$=0.3) | 64.57 | 0.5949 | 0.4924 | 2.4664 |
| + ORNet ($\lambda$=0.5) | 69.18 | 0.6189 | 0.5905 | 2.9944 |
| + ORNet ($\lambda$=0.7) | 69.76 | 0.6352 | 0.6104 | 3.2444 |
| + ORNet ($\lambda$=0.9) | 69.76 | 0.6440 | 0.6198 | 3.3953 |
| + ORNet_4x ($\lambda$=0.1) | 47.86 | 0.5469 | 0.3534 | 1.6511 |
| + ORNet_4x ($\lambda$=0.3) | 62.81 | 0.5509 | 0.5069 | 2.1908 |
| + ORNet_4x ($\lambda$=0.5) | 67.53 | 0.5792 | 0.5738 | 2.6217 |
| + ORNet_4x ($\lambda$=0.7) | 68.15 | 0.5970 | 0.5775 | 2.7956 |
| + ORNet_4x ($\lambda$=0.9) | 68.20 | 0.6051 | 0.5773 | 2.8558 |

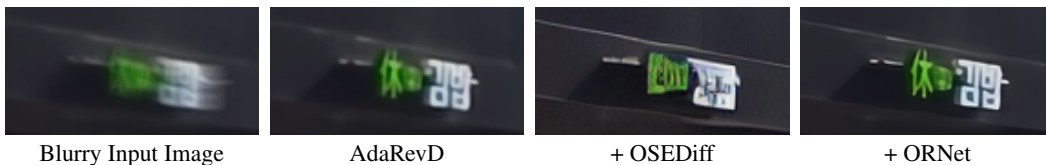

| Blurry Input Image | AdaRevD | + OSEDiff | + ORNet |
|---|---|---|---|

Figure S.8: Comparison of applying the ISR model (OSEDiff) directly and using our ORNet to refine the output of the restoration model (AdaRevD). The ISR model generates details that are not present in the input image, leading to unrealistic results.

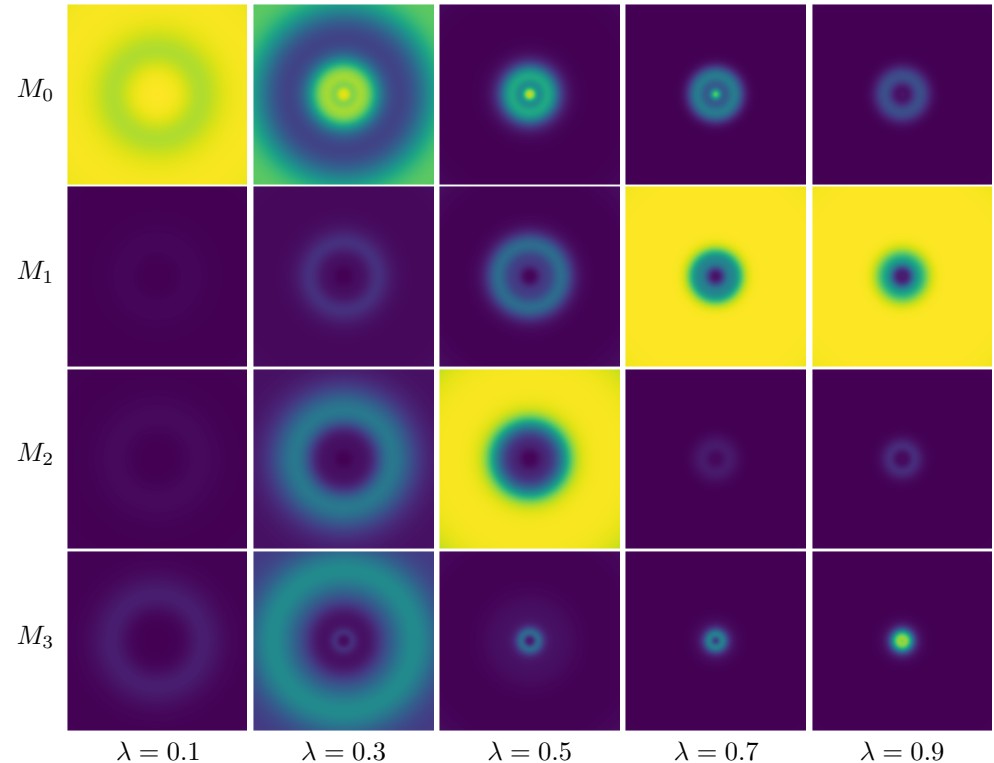

Figure S.9: Frequency masks generated by the proposed controllable frequency mask generator. The enhancement level is controlled by the input parameter $\lambda$.

