# OpenReview forum: "Beyond the Ground Truth: Enhanced Supervision for Image Restoration"
_ICLR.cc/2026/Conference — ICLR 2026 Conference Withdrawn Submission_

### Official Review · Reviewer_LEQ3 · 2025-10-31

**Soundness:** 2
**Presentation:** 3
**Contribution:** 3
**Rating:** 4
**Confidence:** 5

**Summary:**

The paper proposes a novel framework to enhance ground-truth images for improving the supervision quality in real-world image restoration tasks. Since the performance of restoration models is often constrained by imperfect ground-truth data, the authors generate perceptually enhanced ground-truth variants via super-resolution and introduce a conditional frequency mask generator to produce adaptive masks that fuse frequency components from both the original and super-resolved images. This frequency-domain mixup maintains semantic consistency while enriching perceptual details and avoiding hallucinated artifacts. The enhanced ground truths are then used to train a lightweight output refinement network, which can be easily integrated into existing restoration pipelines.

**Strengths:**

1. The idea presented in this paper is interesting.

**Weaknesses:**

1. The description of the Conditional Frequency Mask Generator is unclear. The role and mechanism of λ are not well explained. Based on my observation, it seems that only the network g is trained, yet its training behavior and the optimization objective of the Mask Coefficients remain ambiguous. This section is difficult to follow and requires clearer explanations.

2. As far as I know, super-resolution methods tend to generate fake details and artifacts. If the original ground truth (GT) images are already of high quality, the generated enhanced GT may even degrade in quality. The authors should analyze such cases and conduct ablation studies and analyses on the GT generation process, which are currently missing in the paper.

3. The current experiments are conducted only on deblurring and denoising tasks. To more comprehensively validate the effectiveness of the proposed method, experiments should also be performed on additional restoration tasks, such as deraining and dehazing.

4. The paper lacks a discussion of its limitations.

**Questions:**

no

---

### Official Review · Reviewer_1pSx · 2025-10-31

**Soundness:** 1
**Presentation:** 3
**Contribution:** 2
**Rating:** 2
**Confidence:** 4

**Summary:**

This paper introduces a novel method for image restoration that employs a Diffusion network to produce high-quality ground truth data, which is then used as a new label to train a lightweight restoration network. However, it lacks sufficient validation to demonstrate the effectiveness of each component and to compare against related work.

**Strengths:**

The idea is straightforward, intuitive, and logically presented. The method performs well on the provided datasets and clearly explains each major component, including the model architecture, training procedure, and overall approach.

**Weaknesses:**

My primary concern is that the paper’s content is too limited and does not even include an ablation study. The main contribution can be divided into two parts: (i) using an existing Diffusion model to generate high-quality images (new ground truths), and (ii) employing a lightweight fusion model to refine these results. However, there is no ablation study to validate the contribution of each component. It is already well-known that a Diffusion model can generate high-quality images from the originals; thus, it remains unclear why a simple modulator would not suffice. Moreover, frequency-based operations are not new in this domain. Although the paper discusses their effectiveness, it provides no experimental evidence or comparison with related methods. As a result, the source of the reported strong performance is unclear. Overall, the work lacks sufficient experimental validation to substantiate its approach.

**Questions:**

1. Include an ablation study to demonstrate the contribution of each component (Diffusion model and fusion network).
2. Justify why the Diffusion output alone or a simpler modulator is insufficient.
3. Clarify how the frequency-based operation differs from prior work and provide experiments validating its effectiveness.
4. Explain which component contributes most to the reported performance gains.
5. Add comparisons with recent state-of-the-art image restoration methods.

---

### Official Review · Reviewer_JwzM · 2025-11-01

**Soundness:** 2
**Presentation:** 3
**Contribution:** 2
**Rating:** 4
**Confidence:** 5

**Summary:**

This paper introduces a supervision enhancement framework that combines the original ground-truth (GT) image with additional images generated by a super-resolution (SR) model. The two GT images are fused using the proposed frequency mask-based method. Based on the newly constructed GTs, the authors train a refinement network built upon existing image restoration frameworks. The proposed method improves performance on both in-distribution and out-of-distribution tasks.

**Strengths:**

1. A frequency-domain mixup strategy is proposed to fuse the original GT image with its super-resolved variants, thereby improving the quality of the supervision signals.

2. A refinement network is trained on the enhanced GTs, boosting the performance of existing image restoration frameworks.

**Weaknesses:**

1. Essentially, the authors employ an advanced model to enhance the existing low-quality GT data, generating higher-quality GTs. These generated GTs are then used to train or fine-tune existing image restoration algorithms with the proposed refinement network. However, the novelty of this strategy appears somewhat limited, as the overall performance is largely dependent on the capacity of the advanced model employed. Moreover, the paper does not discuss the criteria or methodology used for selecting this advanced model.

2. The paper does not clearly describe how the parameter $\lambda$ is processed by the model. In addition, it is sensitive to different tasks/baselines.

3. The coefficient prediction network and ORNet are trained jointly on both deblurring and denoising datasets, whereas the competing methods are trained on datasets corresponding to a single degradation type. This raises concerns regarding the fairness of the comparison.

4. The observed performance improvement appears to stem primarily from the capability of the SR model used for GT generation. This may explain why ORNet achieves only marginal PSNR gains in Tables 3 and 4.

**Questions:**

The predefined masks are constructed using a Gaussian basis, whereas the fusion of different GT images is performed in the frequency domain using FFT. Would this introduce a potential inconsistency

---

### Official Review · Reviewer_TjtB · 2025-11-01

**Soundness:** 3
**Presentation:** 3
**Contribution:** 2
**Rating:** 4
**Confidence:** 4

**Summary:**

The paper tackles a practical but under-discussed bottleneck in real-world image restoration (IR): imperfect ground-truth (GT) images in common datasets (e.g., slight blur from video-frame selection in GoPro, averaging-induced blur in SIDD). It proposes a supervision enhancement pipeline that (1) generates perceptually improved GT variants via a one-step diffusion super-resolution model (OSEDiff), then (2) performs frequency-domain mixup using a conditional frequency mask generator to adaptively fuse frequency components from the original GT and its SR variants into an enhanced GT. Building on these enhanced targets, the authors train a lightweight, model-agnostic Output Refinement Network (ORNet) that sits after an existing restoration model and improves perceptual quality under a user-controlled knob λ. They report strong gains on no-reference and VLM-based metrics, OOD robustness, a 70-person user study, and favorable efficiency (≈4.5M params / 20 GMACs). Code, models, and enhanced images are promised for release.

**Strengths:**

1. Motivation is reasonable, which targets the inherent imperfections of GT in existing datasets (e.g.,GoPro; SIDD causing blur), which is worth addressing.

2. The proposed ORNet is lightweight and architecture-agnostic.

3. The experiments are fairly comprehensive: spanning multiple backbones and tasks on GoPro/SIDD, integrating ID/OOD evaluations, no-reference and VLM-IQA metrics, and a 70-participant user study, which together provide a thorough validation of the method’s effectiveness.

**Weaknesses:**

1.  Enhancement strategy (upsample + OSEDiff + downsample).

    (1)  Could this pipeline inject the bias of the diffusion prior into the generated image domain rather than reflecting the real image domain?

    (2)  Can you provide a user study and/or PSNR against GT to demonstrate that this enhancement strategy preserves real-scene details?

    (3)  How do you verify that the enhanced GT does not introduce diffusion-induced hallucinations or artifacts?

2.  Insufficient ablations for ORNet.

    (1)  Please add the upper bound by retraining/fine-tuning the backbone on the enhanced GT directly.

    (2)  Compare simple averaging fusion of $IiGTI\_i^{GT}IiGT$ versus your frequency mixup (an ablation replacing freq-mix with average fusion).

    (3)  Provide ablations on hyperparameters λ and B.

3.  Is the release of enhanced GT compatible with the licenses of the original datasets?

4.  Metrics and baseline selection.

    (1)  For in-distribution (ID) evaluation, you only report IQA metrics and VLM-based scores. Shouldn’t you also report PSNR/SSIM w\.r.t. the enhanced GT?

    (2)  In Tables 3 and 4, why is FFTFormer the only baseline? Can your method work with more recent baselines, e.g., LoFormer (<https://arxiv.org/abs/2407.16993>)?

5.  Comparison with latest SOTA (gyro-based deblurring).
    Could you provide comparisons with Gyro-based Neural Single Image Deblurring (GyroDeblurNet):

    (1)  Direct testing on GyroBlur-Synth and GyroBlur-Real-Synth;

    (2)  Evaluate whether their model + ORNet yields effective improvements.

**Questions:**

See Weaknesses.

---

### Note · Authors · 2025-11-14

I have read and agree with the venue's withdrawal policy on behalf of myself and my co-authors.